# Brain Mechanisms of COVID-19-Sleep Disorders

**DOI:** 10.3390/ijms22136917

**Published:** 2021-06-28

**Authors:** Oxana Semyachkina-Glushkovskaya, Aysel Mamedova, Valeria Vinnik, Maria Klimova, Elena Saranceva, Vasily Ageev, Tingting Yu, Dan Zhu, Thomas Penzel, Jürgen Kurths

**Affiliations:** 1Institute of Physics, Humboldt University, Newtonstrasse 15, 12489 Berlin, Germany; juergen.kurths@pik-potsdam.de; 2Department of Biology, Saratov State University, Atrakhanskaya Str. 83, 410012 Saratov, Russia; mamedovaysel95@gmail.com (A.M.); ler.vinnick2012@yandex.ru (V.V.); mari-1997@mail.ru (M.K.); sophora68@mail.ru (E.S.); old-lon@yandex.com (V.A.); 3Wuhan National Laboratory for Optoelectronics, Britton Chance Center for Biomedical Photonics, Huazhong University of Science and Technology, Wuhan 430074, China; yutingting@hust.edu.cn (T.Y.); dawnzh@mail.hust.edu.cn (D.Z.); 4MoE Key Laboratory for Biomedical Photonics, Collaborative Innovation Center for Biomedical Engineering, School of Engineering Sciences, Huazhong University of Science and Technology, Wuhan 430074, China; 5Sleep Medicine Center, Charité-Universitätsmedizin Berlin, Charitéplatz 1, 10117 Berlin, Germany; 6Potsdam Institute for Climate Impact Research, Telegrafenberg A31, 14473 Potsdam, Germany

**Keywords:** COVID-19-sleep disorders, brain mechanisms, the blood–brain barrier permeability

## Abstract

2020 and 2021 have been unprecedented years due to the rapid spread of the modified severe acute respiratory syndrome coronavirus around the world. The coronavirus disease 2019 (COVID-19) causes atypical infiltrated pneumonia with many neurological symptoms, and major sleep changes. The exposure of people to stress, such as social confinement and changes in daily routines, is accompanied by various sleep disturbances, known as ‘coronasomnia’ phenomenon. Sleep disorders induce neuroinflammation, which promotes the blood–brain barrier (BBB) disruption and entry of antigens and inflammatory factors into the brain. Here, we review findings and trends in sleep research in 2020–2021, demonstrating how COVID-19 and sleep disorders can induce BBB leakage via neuroinflammation, which might contribute to the ‘coronasomnia’ phenomenon. The new studies suggest that the control of sleep hygiene and quality should be incorporated into the rehabilitation of COVID-19 patients. We also discuss perspective strategies for the prevention of COVID-19-related BBB disorders. We demonstrate that sleep might be a novel biomarker of BBB leakage, and the analysis of sleep EEG patterns can be a breakthrough non-invasive technology for diagnosis of the COVID-19-caused BBB disruption.

## 1. The ‘Coronasomnia’ Phenomenon

2020 and 2021 have been unprecedented years due to the modified severe acute respiratory syndrome (SARS) coronavirus (SARS-CoV-2), which spreads rapidly to all continents, leading to a pandemic. The number of cases of coronavirus disease 2019 (COVID-19) continues to rise globally, and between 31 December 2019 and 29 April 2021, 148,999,876 people worldwide have been infected with 3,140,115 deaths (WHO, 2021: https://covid19.who.int, accessed on 1 May 2021). The COVID-19 pandemic has placed an enormous burden on the global healthcare system and the entire population in every country is at risk of infection [1]. SARS-CoV is a novel virus that caused the first major pandemic of the new millennium in 2003 [2,3,4,5,6].

Nowadays, the modified COVID-19 causes atypical infiltrated pneumonia with many neurological symptoms and sleep disorders [7,8,9]. The current COVID-19 pandemic significantly affects our daily activities due to social distancing, which is impacting work, social, educational, and entertainment activities [10,11,12]. Indeed, for the last 1.5 years, the international news is focused mainly on health, employment, finance, and economic recession associated with the COVID-19 pandemic. These uncertainties have exposed many people to stress, such as social confinement, limited daylight exposure, and changes in daily routines, which are also associated with various sleep disturbances, known as the ‘coronasomnia’ phenomenon (Figure 1) [11,13,14]. Notice that it is not a new phenomenon. Several epidemiological studies reported that insufficient sleep, poor sleep quality, insomnia, sleep apnea, and disturbances of sleep–wake schedules are typical manifestations of physical and emotional morbidity in a worldwide pandemic [15]. Several studies have previously documented the impact on sleep of major events, such as earthquakes, floods, wild fires, and war-time [16,17].

The first studies of COVID-19-associated sleep disorders were published in China. Huang and Zhao collected information from a survey of 7236 volunteers and reported 18% of poor sleep quality [10]. Later, the sleep disorders in patients with COVID-19 have also been highlighted in several other publications from different countries [9,18,19,20,21,22,23,24,25,26,27,28,29]. These studies examined the effect on sleep of SARS-CoV-2 infection and confounders related to isolation, quarantine, anxiety, stress, or financial losses. In Europe, symptoms of insomnia could be related to psychosocial factors, such as the confinements due to quarantine [19]. In Italy, COVID-19-related anxiety was highly associated with sleep disorders. In a survey of 2291 Italians, 57.1% reported poor sleep quality [20]. In the International COVID-19 Sleep Study, COVID-19-mediated insomnia, nightmares, sleep apnoea, fatigue, exhaustion, and rapid eye moved (REM) sleep behavior disorder have shown [15]. There is the hypothesis that sleepiness and REM sleep behavior disorder might be related to COVID-19 per se, whereas insomnia might be related mainly to confinement, anxiety, and other psychosocial factors [16].

We found 226 relevant publications out of 652 papers on COVID-19 and sleep disorders until 30 April 2021 in PubMed. Many of these publications are reviews or clinical studies among healthcare workers. Surveys have been completed in the general public in China [10,29], France [30], Italy [11], and USA [31]. Insomnia and other sleep disorders were frequently reported, and there were associations with confinement, physical exercise, social interactions, and other aspects [14]. A recent 1-month cross-sectional observational study of 180 medical staff in China who were treating patients with COVID-19 during January and February 2020 found higher levels of anxiety associated with different types of sleep disturbances [32].

The COVID-19-sleep disorders are a major problem of rehabilitation of patients after infection. However, the exploration of mechanisms underlying the ‘coronasomnia’ phenomenon is still in its infancy. Most research related to COVID-19-sleep disorders is obtained on the effect of lookdown or confinement during the pandemic. There is even a meta-analysis on the many studies published [33] and one case report [34]. However, if we want to know the association of a PCR diagnosed COVID-19 infection and sleep problem, then there are no studies until now. Here, we review important findings and trends in sleep research in 2020–2021, demonstrating how sleep disorders can open the blood–brain barrier (BBB) for inflammatory factors, which might contribute to the ‘coronasomnia’ phenomenon. We also highlight the mechanisms of COVID-19-induced BBB disruption and neuroinflammation that can be an additional reason for COVID-19-related sleep disorders. Finally, we discuss new perspective strategies for the prevention of COVID-19-related BBB disorders. We demonstrate that sleep might be a novel biomarker of the BBB leakage, and the analysis of EEG sleep patterns can be a breakthrough non-invasive technology for the diagnosis of COVID-19-caused BBB disruption.

## 2. Sleep Loss-Associated Neuroinflammation and the Blood–Brain Barrier Disruption

Sleep deprivation in duration and/or quality is a common type of COVID-19-related sleep disorder [12,19,20,21,22,23,24,25,26,27,28,29,30]. We do not know how sleep restriction affects the brain. However, there is emerging evidence that the major complication of sleep loss is neuroinflammation, which induces BBB disruption (Table 1). Indeed, sleep loss per se, including sleep deprivation, sleep restriction, sleep fragmentation, or sleep apnea in human and rodents induces a systemic low-grade inflammation characterized by the release of several molecules, such as cytokines, chemokines, and acute-phase proteins; all of them can promote changes in cellular components of the BBB, particularly on brain endothelial cells [35,36,37,38,39,40,41,42,43,44,45,46]. In this session, we discuss the role of inflammatory mediators that increase during sleep loss and how these changes may alter the BBB permeability as well as we analyze hypothetical mechanisms by which sleep deprivation may induce the BBB disruption, emphasizing the regulatory effect of inflammatory molecules on tight junction proteins.

The first results on sleep deprivation and brain inflammation were published in the early 2000s [47,48]. These findings report that the low-grade systemic inflammation induced by sleep loss is characterized by a subtle but sustained increase in peripheral levels of pro- and inflammatory mediators. Further experimental studies demonstrated that acute and chronic sleep loss is associated with an increased level of proinflammatory mediators, such as tumor necrosis factor-α (TFN-α), interleukin-1β (IL-1β), IL-6, IL-17A, and C-reactive protein (CRP) as well as with an increased level of immune-derived inflammatory mediators, such as cyclooxygenase-2 (COX-2), nitric oxide synthase (NOS), endothelin-1 (ET-1), vascular endothelial growth factor (VEGF), and insulin-like growth factor-1 (IGF-1) [35,39,44,49,50,51].

C-reactive protein (CRP) and high-sensitivity C-reactive protein (hs-CRP) are markers of systemic inflammation and may serve as biomarkers of OSA [52,53]. Meta-analysis on the total of 1297 subjects revealed that serum CRP levels in the OSA group were 1.98 mmol/L higher than those in the control group (95% confidence interval: 1.39–2.58, *p* < 0.01). Similarly, serum hs-CRP levels in the OSA group were 1.57 mmol/L higher than that in the control group (95% confidence interval: 0.96–2.18, *p* < 0.01) [52].

Recently, it has been discovered that sleep deprivation increases the BBB permeability to inflammatory mediators, immune cells, and exogenous tracers in humans and rodents [35,36,38,39,41,42,43,45,50,54,55,56].
ijms-22-06917-t001_Table 1Table 1The effects of sleep deprivation on the blood–brain barrier functions.ReferencesSleep Loss ModelsThe Effects of Sleep Loss on the BBB PermeabilityMedina-Flores F. et al. (2020) [38]Wistar rats were sleep-restricted 20 h daily with 4 h sleep recovery for 10 days.Sleep loss disrupts pericyte-brain endothelial cell interactions.Hurtado-Alvarado G. et al. (2016) [39].ReviewAcute and chronic sleep deprivation, sleep restriction and sleep fragmentation.Sleep loss induces a low-grade systemic inflammation characterized by the release of several molecules, such as cytokines, chemokines, and acute-phase proteins; all of them may promote changes in cellular components of BBB, particularly on brain endothelial cells.Hurtado-Alvarado G. et al. (2018) [35]Mice were sleep-restricted during 10 days using the flowerpot technique for 20 h per day with 4 h of daily sleep opportunity.Cytokines may play a key role in modulating BBB function during sleep restriction via the overexpression of Iba-1, MMP-9 and A2A adenosine receptors.He J. et al. (2014) [50]Chronic sleep restriction of mice for 6 days in a rotatory bar for 12 h per day. Sleep restriction of this method induced REM sleep loss in the first3 days with partial REM sleep recovery afterward.Chronic sleep restriction diminished endothelial and inducible nitric oxide synthase, endothelin1, and glucose transporter expression in cerebral microvessels of BBB and decreased 2-deoxy-glucose uptake by the brain. The expression of several tight junction proteins was decreased, whereas the level of cyclooxygenase-2 increased. This coincided with an increase of paracellular permeability of BBB to the small tracers sodium fluorescein and biotin. Chronic sleep restriction for 6 d was sufficient to impair BBB structure and function, although the increase of paracellular permeability returned to baseline after 24 h of recovery sleep.Hurtado-Alvarado G. et al. (2017) [56]Male Wistar rats were sleep restricted using the modified multiple platform method for 10 days, with a daily schedule of 20-h sleep deprivation plus 4-h sleep recovery at their home-cages.Chronic sleep restriction disrupts interendothelial tight junctions in the hippocampus and increases BBB permeability to fluorescein-sodium, and decreases interendothelial junction complexity by increasing the frequency of less mature end-to-end and simply overlap junctions, even after sleep recovery, as compared to intact controls. Chronic sleep loss also induces the formation of clefts between narrow zones of adjacent endothelial cell membranes in the hippocampus.Gómez-González B. (2013) [55]REM sleep restriction was induced by the multiple platform technique; male rats were REM sleep-restricted 20 h daily (with 4 h sleep opportunity) for 10 days; control groups included large platform and intact rats.REM sleep restriction increased BBB permeability to Evans blue in the whole brain. Brief periods of sleep recovery rapidly and effectively restored the severe alteration of BBB function by reducing BBB transfer of Evans blue. The mechanism of BBB breakdown involved increased caveolae formation at brain endothelial cells. REM sleep regulates the physical barrier properties of BBB.Hurtado-Alvarado G. et al. (2016) [54]The chronic sleep restriction of male Wistar rats during 10 days.Sleep restriction increased BBB permeability to FITC-dextrans and Evans blue, and the effect was reverted by the administration of selective A2A adenosine receptor antagonist (SCH58261) in almost all brain regions excluding the cerebellum. Sleep restriction increased the expression of A2A adenosine receptor only in the hippocampus and basal nuclei without changing the expression of adenosine-synthesizing enzyme (CD73) in all brain regions. Sleep restriction reduced the expression of tight junction proteins (claudin-5, occludin, ZO-1) in all brain regions, except in the cerebellum; and SCH58261 restored the levels of tight junction proteins in the cortex, hippocampus and basal nuclei. Sleep restriction-induced neuroinflammatory markers (GFAP and Iba-1) overexpression that was attenuated with the administration of SCH58261.Daulatzai M.A. (2016) [42]. ReviewObstructive sleep apneaObstructive sleep apnea is a risk factor triggering neuroinflammation and oxidative-nitrosative stress that in turn decrease nitric oxide and enhance endothelin, amyloid-β deposition, cerebral amyloid angiopathy, and BBB disruption.Lim D.C. and Pack A.I. (2014) [41]. ReviewObstructive sleep apneaCyclical intermittent hypoxia is a stressor that disrupts BBB via molecular responses already known to occur in either obstructive sleep apnea patients or animal models of intermittent hypoxia.Voirin A.C. et al. (2020) [43]Obstructive sleep apnea. The two groups of volunteers were selected, a group of patients suffering newly diagnosed severe obstructive sleep apnea (AHI > 30/h) and a group showing no sleep apnea (AHI <5/h). The human in vitro BBB model of endothelial cells (HBEC-5i) with sera of patients with and without obstructive sleep apnea was studied.After incubation with sera from patients with obstructive sleep apnea, there was a loss of integrity in the human in vitro BBB model; this was reflected by an increase in permeability (43%; *p* < 0.001) and correlated with a 50% and 40% decrease in tight junction protein expression of ZO-1 and claudin-5, respectively. There was an upregulation in Pgp protein expression (52%) and functionality and a downregulation in BCRP expression (52%). These results demonstrate that severe BBB disorder after exposure to sera from patients with obstructive sleep apnea was reflected by BBB opening.Benedict C. et al. (2014) [45]Healthy young men were divided on 2 groups including either 8-h of nocturnal sleep [22:30–06:30] and total sleep deprivation.Acute sleep deprivation increases serum levels of neuron-specific enolase (NSE) and S100 calcium-binding protein B (S-100B) in healthy young men.Opp M.R. et al. (2015) [46]The sleep fragmentation device used in this study. The device consists of a cylindrical Plexiglass^®^ chamber divided into two separate compartments. The floor of the chamber is a disc that is programmed to rotate at specific intervals. In this study, disc rotations were confined to the 12 h light period, and consisted of an 8 s rotation once every 30 s, on average. The direction of the disc rotation was randomized, and the precise placement of the 8 s rotation within the 30 s period was varied to prevent behavioral adaptation by a mouse to disc movements. Sleeping mice wake up when the disc rotates, whereas awake mice continue their normal behavior. During the 12 h dark period, there was no disc rotation, and mice were allowed spontaneous behavior. Mice were housed in the sleep disruption devices, one mouse in each compartment, for 3 days of habituation prior to beginning the sleep fragmentation protocol.The sleep fragmentation disrupted the BBB and increased tumor necrosis factor-α transport in aged mice but not in young mice.Pan W. and Kastin A.J. (2017) [36]. ReviewAcute and chronic sleep deprivation, sleep restriction, and sleep fragmentation.Authors summarize research on the sleep-BBB interactions in five sections: (1) the structural basis enabling the BBB to serve as a huge regulatory interface; (2) BBB transport and permeation of substances participating in sleep-wake regulation; (3) the circadian rhythm of BBB function; (4) the effect of experimental sleep disruption maneuvers on BBB activities, including regional heterogeneity, possible threshold effect, and reversibility; and (5) implications of sleep disruption-induced BBB dysfunction in neurodegeneration and CNS autoimmune diseases.

Hurtado-Alvarado et al. observed that sleep restricted C57BL/6 mice exhibited an increase in the BBB permeability to Na-fluorescein, 10 kDa dextrans, and Evans blue in comparison to the same strain control groups. Moreover, there were differences in the distribution of the tracers in the brain [35]. Na-fluorescein and 10 kDa dextrans, reflecting the BBB leakage to low molecular weight [57], was presented a widespread deposition in the parenchyma of the cortex, hippocampus, and vermis in C57BL/6 sleep-restricted mice [35]. In contrast, the larger tracer, Evans blue 65.8 kDa, mimicking the BBB breakdown to proteins [57], was presented mostly in the hippocampus.

Gómez-González et al. 2013 reported that 10 days of sleep restriction in a schedule of 20 daily hours of sleep deprivation using the multiple platform technique plus 4 h of sleep opportunity induce a full suppression of REM sleep and around 30% loss of non-REM sleep [55]. This finding showed a widespread breakdown of the BBB. Interestingly, the brief periods of sleep opportunity (40 to 120 min) induced a progressive recovery of the BBB permeability to Evans blue (65.8 kDa) in the majority of brain regions studied, with the exception of the hippocampus and the cerebellum. In a subsequent study, mice were subjected to sleep restriction for 6 days in a rotatory bar for 12 h per day. Sleep restriction induced REM sleep deprivation in the first 3 days with partial REM sleep recovery afterward [50]. Under these conditions, the increased BBB permeability to low molecular-weight tracer—sodium fluorescein (10 kDa) and the decreased mRNA levels of the tight junction proteins claudin-5, zonula occludens-2, and occludin was observed in all regions of the brain. It is important to note that the sleep recovery by 24 h fully reverted the BBB permeability in mice [50]. 

Sleep disturbances also increase the BBB permeability in humans [41,42,43,45]. Indeed, acute sleep deprivation induces an increase in the serum levels of neuron-specific enolase and S100 calcium-binding protein B (S-100B) in healthy young men [45]. Obstructive sleep apnea is a risk factor that triggers neuroinflammation and oxidative-nitrosative stress. That in turn, decreases nitric oxide and enhances endothelin, amyloid-β deposition, cerebral amyloid angiopathy, and BBB disruption [42]. Voirin A.C. et al. used the human in vitro BBB model of endothelial cells (HBEC-5i) with sera of patients with and without obstructive sleep apnea [43]. They demonstrated that obstructive sleep apnea is accompanied by BBB disruption and a decrease in tight junction protein expression of zonula occludes-1 and claudin-5. 

There are age-related differences in the BBB dysfunction induced by sleep loss. Indeed, sleep fragmentation disrupts the BBB and increases the tumor necrosis factor-α transport in elderly mice but not in young mice [47].

Previous reports have shown that the mechanism of BBB opening mediated by sleep loss involves both tight junction disentangle and the increase of pinocytosis in the cerebral endothelial cells [50,55,56]. Medina-Flores F et al. using a model of chronic sleep loss during 10 days showed that sleep deprivation disrupts pericyte-brain endothelial cell interactions [38]. Chronic sleep restriction disrupts interendothelial tight junctions in the hippocampus, causes a BBB leakage to fluorescein-sodium and decreases the interendothelial junction complexity. Chronic sleep deprivation also induces the formation of clefts between narrow zones of adjacent endothelial cell membranes in the hippocampus [56]. 

Chronic sleep restriction causes an increase of paracellular permeability of the BBB to the small tracers sodium fluorescein and biotin that is accompanied by reducing the expression of endothelial and inducible NOS, endothelin1, glucose transporter in cerebral endothelium and decreases 2-deoxy-glucose uptake by the brain [50]. The expression of several tight junction proteins also decreases, whereas the level of cyclooxygenase-2 increases. Chronic sleep restriction for 6 d is sufficient to impair the BBB structure and function, although the increase of paracellular permeability returns to baseline after 24 h of recovery sleep [50]. 

Cytokines may play a key role in modulating the BBB function during sleep restriction via the overexpression of the tumor necrosis factor α [48,58], interleukin-1 beta [59], interleukin-6 [48,60], interleukin-17 [61], C reactive protein [61,62], allograft inflammatory factor 1 (Iba-1) [33], matrix metalloproteinase 9 (MMP-9) [35] and A2A adenosine receptors [35]. There are two reviews discussing that sleep deprivation induces a low-grade systemic inflammation characterized by the release of inflammatory molecules, such as cytokines, chemokines, and acute-phase proteins, which promote changes in cellular components of the BBB [36,39]. The administration of the selective A2A adenosine receptor antagonist (SCH58261) protects BBB injuries after sleep restriction in almost all brain regions, excluding the cerebellum [54]. Sleep restriction reduces the expression of tight junction proteins (claudin-5, occludin, zonula occludin-1) in all brain regions, except in the cerebellum; and SCH58261 restores the levels of tight junction proteins in the cortex, hippocampus and basal nuclei. Sleep restriction induces neuroinflammatory markers (GFAP and Iba-1) overexpression that was attenuated with the administration of SCH58261 [54].

Overall, sleep loss causes a low-grade inflammation of the brain, which facilitates the BBB breakdown that can partly explain the ‘coronasomnia’ phenomenon. Indeed, COVID-19-related sleep disorders might contribute to the BBB opening that can be a door for COVID-19 entry into the brain and further hyper-inflammation of the CNS. Figure 2 illustrates how sleep disorders can open the BBB and contributes SARS-CoV-2 entry into the brain via neuroinflammation induced by sleep deprivation (Figure 2).

## 3. Circadian Regulation of the Blood–Brain Barrier Permeability to Cytokines

The circadian rhythm is a biological clock that endogenously oscillates with a period of roughly 24 h. There is recognition of the importance that the central circadian pacemakers in mammals are both the suprachiasmatic nucleus and the BBB cells, which have autonomous circadian rhythms driven by molecular clocks [37,63,64,65]. Moreover, the large cerebral blood vessels themselves exhibit the circadian clock [66,67]. In the following, we highlight recent evidence demonstrating that the BBB permeability to cytokines depends on the time of day that can be important for a better understanding of the mechanisms underlying COVID-19-associated sleep disorder. 

Upon invasion of bacteria and viruses, the immune system is immediately activated, producing inflammatory mediators, including pro-inflammatory cytokines, such as tumor necrosis factor (TNF-α), interleukins, IL-1β/IL-6, and inflammatory mediators, prostaglandins (PGs) and leukotrienes. This immune response evokes an acute inflammatory process (hours to days) to clear the pathogens from the infected tissues. 

The accumulating evidence suggests that the severity of COVID-19 is associated with an increased level of inflammatory mediators, including cytokines and chemokines such as interleukins, TNF, granulocyte colony-stimulating factor (G-CSF), monocyte chemoattractant protein-1, macrophage inflammatory protein 1 alpha, C-reactive protein, ferritin, and D-dimers in blood upon SARS-CoV-2 infection [68,69,70,71,72]. Note that among the elevated inflammatory mediators, the blood IL-6 level is highly correlated with the disease mortality when COVID-19 survivors and non-survivors are compared [73,74]. This fact suggests that fatal COVID-19 is characterized as a cytokine release syndrome that is induced by a cytokine storm with high mortality [75,76,77].

The entry of some cytokines, such as tumor necrosis factor (TNF-α) [78,79], interleukins 6 (IL 6) [80,81], and IL-1α [82] into the brain undergo significant circadian oscillations (Figure 2).

The TNF-α is a cytokine used by the immune system for cell signaling. When macrophages detect an infection, they release TNF to alert other immune system cells as part of an inflammatory response. Pan et al. showed that the spinal cord, but not the brain, demonstrate a circadian rhythm in the uptake of TNFα [78]. The greatest TNFα uptake occurs between Zeitgeber time (ZT) 20-ZT23 (Figure 3). This pattern is similar to that of leptin but different from that of interleukin-1. The circadian rhythm of the influx of TNFα into this region of the CNS suggests a functional role for the spinal cord in the physiological actions of TNFα. TNFα transport across the BBB is abolished in receptor knockout mice [79].

Interleukin-6 is a cytokine with pleiotropic actions in both the periphery of the body and the CNS. Altered IL-6 secretion has been associated with inflammatory dysregulation and several adverse health consequences [83]. The intense BBB permeability to radioactively labeled murine and human IL-6 occurs in the evening, while the peak of diurnal uptake of circulating 125I (I-IL) by the brain and the spinal cord is observed at 08.00 h–12.00 h (Figure 2) [70,71,72]. 

This diurnal variation in the rate of uptake of cytokines by the CNS could underlie differences in potency when given at different times of the day.

The activation of cytokines critically depends on endogenously produced prostaglandin D2 (PGD2) that exhibits daily oscillations in the rat CSF, peaking at 14:00 [84,85,86,87].

Evidence indicates that noradrenaline (NA) elicits anti-inflammatory actions in the CNS and consequently may play a neuroprotective role where inflammatory events contribute to the CNS pathology [88,89].

A study by Staedtke et al. using subcutaneously implanted osmotic pumps to continuously release NA revealed that epinephrine may fuel the cytokine storms [90]. Luo and Li analyzed 206 patients who were diagnosed with COVID-19 and suffered a rescue treatment with epinephrine at the Tongji Hospital in Wuhan, China, from January 27 to March 1, 2020. They noticed that only 4/206 (2%) of patients were survivors and discharged from the hospital eventually [91]. The severity of the disease may account mostly for the poor outcome. However, as expected from the established role of NA in fueling the cytokine storms, the aggravated cytokine storms may also play a role in the failure of the rescue.

The brain areas with the highest concentration of noradrenergic nerve terminals and NA have a circadian rhythm in their content of NA in rabbits, rats, and cats [92]. The anterior and lateral hypothalamus [93] and cervical spinal cord [94], midbrain [95], and caudate nucleus [96], medial lower brain stem [97], and frontal cortex [98] all have significant circadian variations in their NA content. These brain regions have their highest NA level during the night in rats, a nocturnal animal, and during the day in rabbits and cats [92]. The noradrenergic locus coeruleus neurons fire in a wake-dependent manner; they are highly active during wake, slow-firing during non-REM sleep, and almost completely quiescent during REM sleep [99,100].

Glucocorticoids play an important role in regulation of the circadian clock via modulating the suprachiasmatic nucleus response and synthesis of various systemic factors, including cytokines [101,102]. Circadian control of glucocorticoid production and secretion involves a central pacemaker in the hypothalamus, the suprachiasmatic nucleus, as well as a circadian clock in the adrenal gland itself. The rhythmically released glucocorticoids, in turn, might contribute to the synchronisation of the cell-autonomous clocks in the body and interact with them to time physiological dynamics in their target tissues around the day.

The BBB consists of vascular endothelial cells connected to each other by tight junctions that form a lumen for blood flow. Endothelial cells express a set of transporters to move molecules both into and out of the brain [103,104], which include permeability-glycoprotein multidrug transporter (also known as Pgp, multidrug resistance protein 1, or ATP-binding cassette sub-family B member 1), an ATP-dependent efflux pump that can pump both endogenous molecules and exogenous compounds back into the lumen of blood vessels [105,106]. The surface of endothelial cells exposed to the brain is surrounded by specialized projections (endfeet) of astrocytic glial cells.

Pgp regulates the permeability of many endogenous and exogenous molecules across the BBB in a circadian manner. Indeed, the peak concentration of quinidine, a substrate of Pgp, in the CSF and ISF of rats is higher between ZT0, ZT4, and ZT8 [107]. This suggests that Pgp is more active during periods of wakefulness. Pgp activity is increased during times of wakefulness also in flies [108]. This is due, in part, to active efflux through Pgp-like transporter during the day [108]. Loss of the circadian clock inhibits the nighttime increase in permeability, suggesting that the clock normally inhibits Pgp-like transporters at night to allow increased permeability [108]. Interestingly, genetic inhibition of Pgp transporters in flies led to increased sleep, possibly by increasing central nervous system (CNS) permeation of peripheral sleep-promoting molecules [109].

## 4. The COVID-19 Attack on the Blood–Brain Barrier Integrity

An alternative route of COVID-19 to the CNS is through the dissemination of SARS-CoV-2 into the systemic circulatory system following the infection of the respiratory tract [110]. In this pathway, SARS-CoV-2 may spread to other target tissues and organs with cerebral blood flow. However, the virus cannot simply migrate from the capillaries of the systemic circulatory system to the brain through endothelial cells due to the unique physiology of the BBB. The BBB is a crucial immunological feature of the CNS. Composed of many cell types, the BBB is both a structural and functional roadblock to microorganisms, such as bacteria, fungi, viruses, or parasites. As a result, the BBB is a key regulator of antigens entry into the CNS and exists at the interface of blood vessels and interstitial fluid throughout the brain. 

Thus, to infect the brain via the hematogenous route, the virus must first bypass the BBB. Human angiotensin I-converting enzyme-2 (ACE2) and transmembrane protease serine 2 are two surface membrane receptors that are involved in SARS-CoV-2 entry into host target cells [111,112,113]. Neuropilin-1 (NRP-1), a member of a family of signaling proteins, was shown to serve as an entry factor and potentiate SARS-CoV-2 infectivity [114,115]. Expression of the ACE2 receptor [116] and NRP1 [117] has also been confirmed in human brain microvascular endothelial cells (BMVECs), making them potential targets for SARS-CoV-2 infection. To date, at least one study has described via post-mortem analysis of the brain by transmission electron microscopy the presence of viral-like proteins inside BMVECs in the frontal lobe of a patient who died from COVID-19 [118]. This provides the first direct evidence that SARS-CoV-2 can infect BMVECs of the BBB [119]. 

Limited evidence exists about the effects of other types of coronaviruses on the BBB. Laboratory animal studies have shown that the coronavirus mouse hepatitis virus (MHV) induces the BBB breakdown regulated by a decrease in the tight junction proteins occludin and zonule occludens 1, resulting in a CNS infection [120]. In the owl monkey (Aotus trivirgatus), infection with the murine coronavirus (strain JHM) (MHV-JHM) combined with outer membrane protein gene (omp1) (JHM OMP1) via intracerebral, intranasal, or intravenous inoculation resulted in a CNS infection, as evidenced by the detection of viral products in the brains of all animals, predominantly in the blood vessels and perivascular regions [119,121,122]. This suggests that a coronavirus can infect and replicate in endothelial cells to bypass the BBB. Moreover, in vitro studies later demonstrated that the coronavirus JHM OMP1 can infect cultured BMVECs isolated from humans and rhesus macaques [119,123], providing further evidence that certain coronaviruses can infect BMVECs of the BBB. 

Preliminary results suggest that SARS-CoV-2 can invade BMVECs in humans [118]. In addition, a recent study indicates that the spike protein of SARS-CoV-2 can induce an increase in the BBB permeability in a BBB-on-a-chip in vitro system [124]. Most importantly, emerging evidence from human studies indicates that SARS-CoV-2 induces BBB dysfunction in humans. Bellon et al. recently reported that, from 31 COVID-19 patients with neurological manifestations, 58% presented an increase in the BBB permeability [125], providing the first-in-human evidence that SARS-CoV-2 induces a BBB dysfunction. In this study, it was unclear whether the disruption of the BBB was a direct result of SARS-CoV-2 infection or this damage was a secondary response to neuroinflammation. More research is needed to elucidate the mechanisms by which this virus bypasses the BBB and enters the brain and whether patients with underlying conditions that affect the BBB are at an increased risk of brain infections [119]. 

Human neurotropic RNA viruses have evolved as opportunistic pathogens that can bypass the BBB and gain entry into the CNS by several mechanisms: (a) paracellular transport via a leaky BBB, (b) transcellular transport by direct infection of the cerebrovascular endothelial cells, or (c) transport via extracellular vesicles, a form of «Trojan horse» trafficking. A recent study on the Japanese encephalitis virus (JEV) (an RNA virus) suggests that the paracellular mode of trafficking could be one of the potential routes of entry into the CNS. JEV infected mast cells release chymase, a vasoactive protease, which cleaves TJ proteins, including zona occludens-1 and 2, claudin-5, and occludin, breaking down the BBB and facilitating the entry of JEV into the CNS [126]. During transcellular migration, viruses invade host endothelial cells to cross the BBB. During paracellular migration, viruses invade tight junctions formed by endothelial cells of the BBB [127,128]. During the Trojan horse strategy, a virus is engulfed by phagocytic host cells, such as neutrophils and macrophages. As lymphocytes, granulocytes, and monocytes all express ACE2, the SARS-CoV might be able to infect them [129,130,131,132], and it is likely that SARS-CoV-2 also may act in the same manner. Moreover, the COVID-19-related systemic inflammation would increase the BBB permeability, thus facilitating the invasion of the CNS by the infected immune cells [133,134]. 

The fact that SARS-CoV-2 can infect macrophages, astroglia, and microglia makes it possible for the host’s immune-mediated response to playing a substantial role. In some patients who died because of COVID-19, a multiple organ failure and a hyperinflammatory syndrome (the «cytokine storm») were hypothesized as possible underlying causes [134,135]. The cytokine storm associated with SARS-CoV-2 infection results in an increased secretion of pro-inflammatory cytokines and chemokines such as IL-6, TNF-α, macrophage inflammatory protein 1-alpha, IP-10, and granulocyte-colony stimulating factor as well as C-reactive protein and ferritin. These observations indicate that upregulation of cytokines and systemic inflammation may be linked to disease severity. Cytokines and chemokines can bind to specific receptors on the cerebral microvascular endothelium leading to a BBB breakdown, neuroinflammation, and encephalitis. The loss of BBB integrity could loosen the tight junctions between the endothelial cells paving the way for paracellular traversal of SARS-CoV-2 into the CNS [126]. 

Due to the loss of BBB integrity, the endothelial cells, pericytes, and astrocytes compromise the ability to prevent immune cells from infiltrating the brain. Rather, immune cells are able to permeate the barrier and infiltrate the CNS, possibly attacking the brain cells, including the neurons. The resulting neuroinflammatory process may result in severe damage to brain function, as reported in [128,136,137].

In total, SARS-CoV-2 via the ACE2 receptor on BMVEC cells can make entry into the brain, inducing a cytokine storm, endothelial inflammation, and alter the BBB integrity, which may facilitate COVID-19-related sleep disorders (Figure 4). Thus, anti-cytokine-based therapeutics or therapies that prevent the BBB damage may be effective in treating patients with COVID-19 associated sleep changes. Additional studies that investigate molecular mechanisms that underlie SARS- CoV-2 associated BBB injuries are warranted. 

## 5. The Perspective Strategies for Prevention of COVID-19-Related BBB Disorders

As discussed above, COVID-19-related sleep disorders are strongly associated with BBB leakage. Thus, the development of new technologies for the analysis of the BBB permeability can be a revolutionary step in the prevention of COVID-19-mediated BBB injuries. However, the monitoring of the degree and duration of the BBB permeability is a challenging problem [157,158]. Therefore, the interest in the development of methods for the assessment of BBB leakage in clinical practice has substantially increased over the past few years [157,158]. Computer tomography has been used to receive information about the BBB integrity [159,160], but it has limited value. The dynamic contrast-enhanced magnetic resonance imaging (MRI) has proven valuable in the assessment of brain pathologies, which are accompanied by the BBB opening [157,158]. However, MRI is bulky and cannot be used at the bedside. Moreover, this procedure is expensive and is performed with contrast agents, which can be toxic [161,162]. This limits its continuous application and use, especially in children and patients with kidney pathology [163,164]. Therefore, the development of novel promising real-time, bedside, non-invasive, label-free, economically beneficial, and readily applicable methods is of the highest actual importance.

Here we discuss that the EEG analysis of sleep patterns can be a promising candidate for the BBB analysis and a breakthrough non-invasive technology for diagnosis of COVID-19-caused BBB disruption. 

It is widely accepted that the main function of the BBB is the protection of the CNS from the penetration of microorganisms and toxins from the blood. However, the latest findings changed our understanding of the role of BBB in the keeping of CNS health. The BBB opening by music [165] or light [166] stimulates the lymphatic clearance of macromolecules from the brain. These new findings partly explain the fact that the BBB opening even without pharmacological therapy contributes to the clearance of amyloid-beta from the brain in patients with Alzheimer’s disease and in mouse models of amyloidosis [167,168,169,170]. We do not know the mechanisms underlying the interrelation between sleep and the BBB opening. However, there is strong evidence that both sleep and the BBB are interlinked with activation of clearance of macromolecules and toxins from the brain that can be accompanied by similar changes in the neurological activity of the brain [165,166,171,172,173,174,175]. 

There is the hypothesis that the sleep EEG slow wave activity (SWA) presents a new informative platform for extracting information about the BBB. Indeed, both sleep [171,172,175] and the BBB opening in humans and animals [173,174,176] are characterized by special EEG patterns in SWA. The mechanisms of the EEG pattern changes during the BBB opening remain unknown. An increase in the BBB permeability can affect the EEG activity in different ways.

The BBB opening can directly affect the EEG dynamics. The signals generated by the BBB originate from a transendothelial resistance between blood and brain tissue. This voltage is a membrane potential of endothelial cells forming the BBB [177]. The BBB opening changes the voltage of endothelial cells, causing depolarization of their membranes [178,179]. These changes of cell potential cause up to mV-level shifts in human scalp EEG [176,178,179]. Several experimental [180,181,182] and clinical [179,183,184] studies suggest that the BBB opening is accompanied by specific EEG changes, especially in SWA. 

The BBB opening can influence the EEG behavior indirectly via astrocytes, which are essential for the BBB integrity and the EEG behavior [185,186]. The astrocytic mechanism of EEG changes may be related to the astrocytic-regulation of synaptic conductance [187,188,189], which is crucial for the electrical activity of cortical neurons. 

Although it is too early to argue that EEG sleep patterns are a reliable marker of the BBB opening, it is not unreasonable to view that the EEG analysis of night brain activity can become a promising tool for effective extraction of information about the BBB permeability to SARS-CoV-2, cytokines and other substances. The development of new technologies of an EEG-based analysis of the BBB can be a revolutionary step in the prevention of COVID-19-sleep disorders and neurological diseases associated with BBB dysfunction. We have recently discussed the mechanisms of BBB-related EEG changes and the methods of modulations of these processes in our previous review [190]. 

## 6. Conclusions

To summarize, the COVID-19 pandemic is accompanied by the development of a growing number of COVID-19-related sleep disorders, known as the ‘coronasomnia’ phenomenon. The most dangerous consequence of sleep disturbances is the BBB disruption when the viruses, bacteria, and toxins can enter the brain and cause neuroinflammation. Therefore, the control of sleep hygiene and quality should be incorporated into routine therapy practice of COVID-19 patients. We highlight the pioneering direction in sleep research 2020–2021 and show that the EEG analysis of sleep patterns can be a revolutionary step in the control of COVID-19-caused BBB disruption. Furthermore, we discuss that SARS-CoV-2 can enter into the brain, directly altering the BBB integrity, which may facilitate COVID-19-related sleep disorders (Figure 4). The therapies that prevent the BBB damage may be effective in treating patients with COVID-19 and COVID-19-related sleep disorders. Additional studies that will be focused on the development of breakthrough technologies of an EEG-based analysis of BBB leakage as well as on the study of molecular mechanisms underlying SARS-CoV-2 associated sleep disorders and the BBB injuries are warranted.

## Figures and Tables

**Figure 1 ijms-22-06917-f001:**
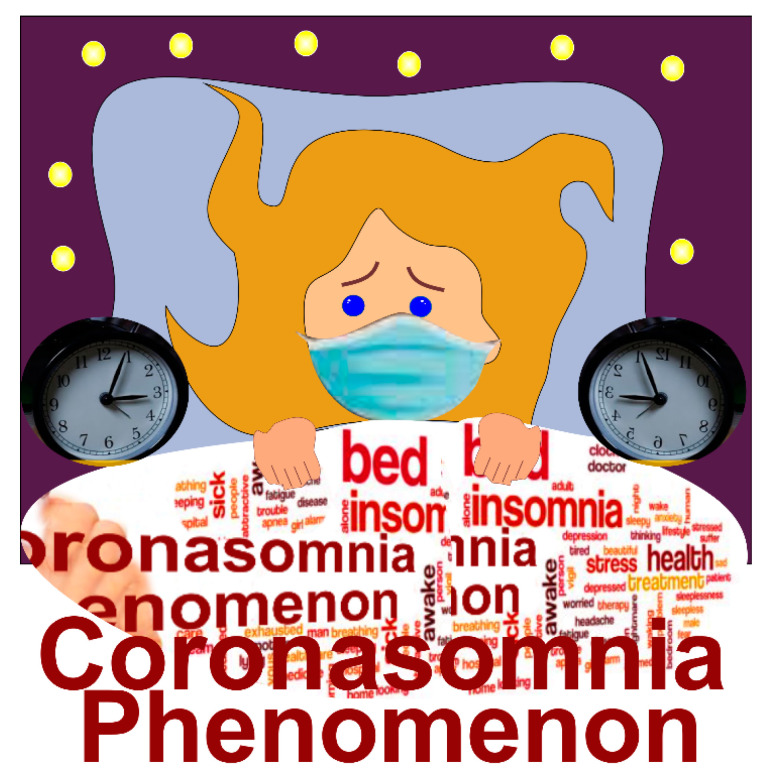
The ‘coronasomnia’ phenomenon has been recently described as COVID-19-related sleep disorders, including insufficient sleep, poor sleep quality, insomnia, and disturbances of sleep-wake schedules associated with physical and emotional morbidity in the COVID-19 pandemic.

**Figure 2 ijms-22-06917-f002:**
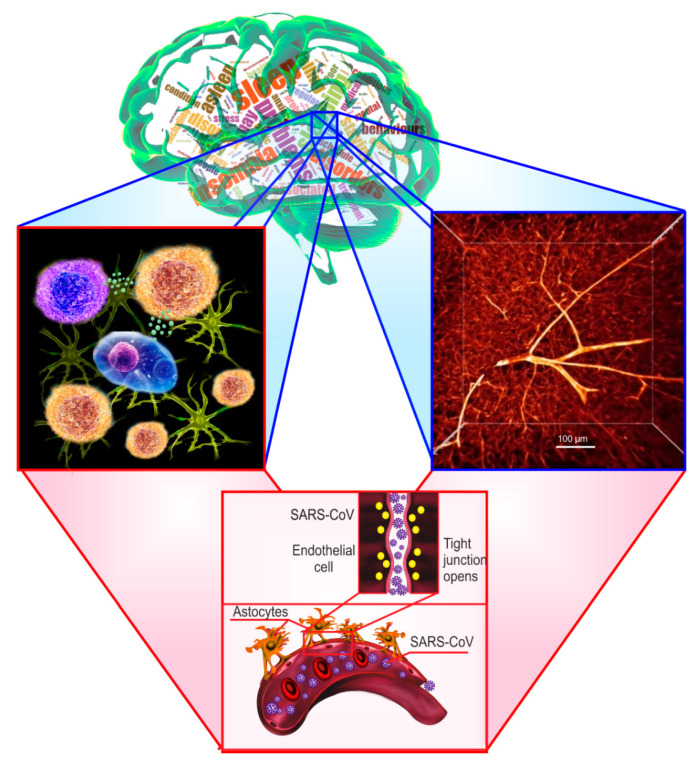
Illustration of the blood–brain barrier (BBB) disruption induced by sleep-related disorders. Middle upper figure demonstrates the various sleep disturbances, including insomnia, sleep deprivation, sleep apnea, associated with the BBB breakdown. The left figure shows that sleep loss causes low-grade inflammation of the brain, which facilitates the BBB disruption. Right (two-photon imaging of the brain capillaries filled by tetramethylrhodamine isothiocyanate–dextran 70 kDa) and the middle lower squares demonstrate the cerebral blood vessels damaged by cytokines leading to the BBB opening.

**Figure 3 ijms-22-06917-f003:**
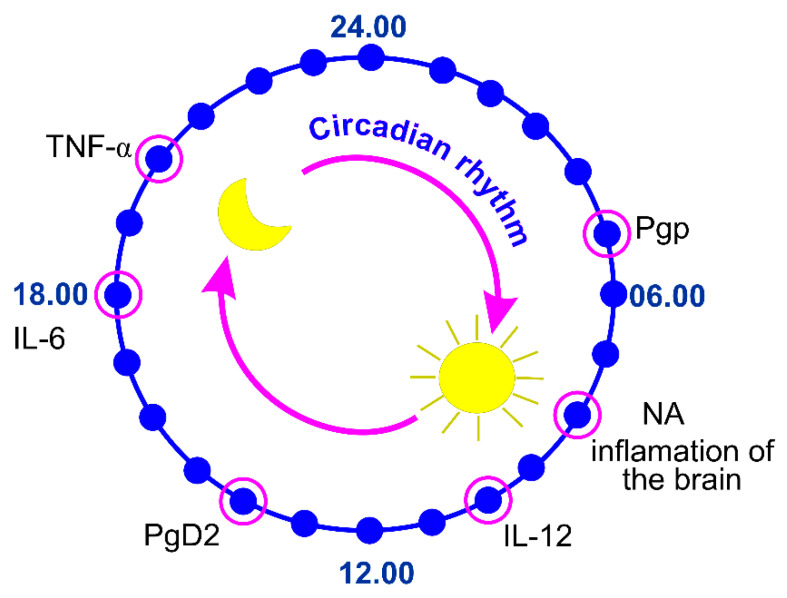
The biological clock of the blood–brain barrier permeability to cytokines (TNF-α, IL-6, IL-1α) and regulating their activity factors, such as hormones (NA and PGD2) and permeability-glycoprotein multidrug transporter (Pgp).

**Figure 4 ijms-22-06917-f004:**
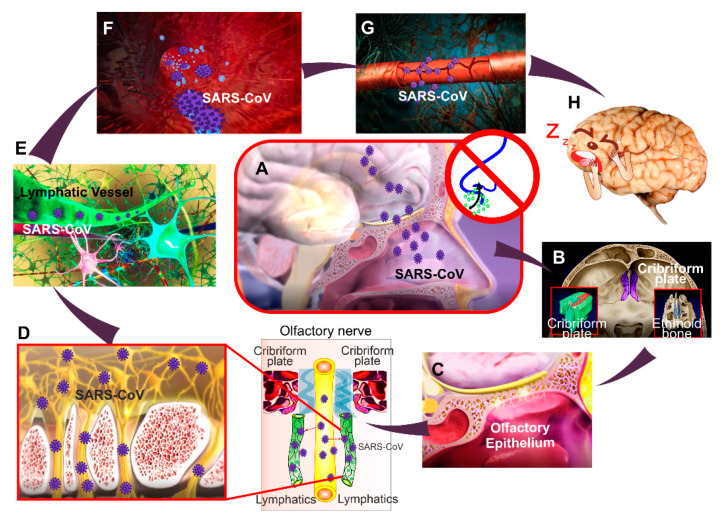
Brain mechanisms of COVID-19-sleep disorders: (**A**–**C**) The one hypothesis explains that the SARS-CoV-2 infiltrates the brain (**A**), possibly from the cribriform plate (**B**) and the olfactory epithelium (**C**). The cribriform plate (**B**) is a center of the lymphatic pathway of metabolic clearance and is a connective bridge between the cerebral spinal fluid and the cervical lymphatic system) [138,139,140,141,142,143,144,145]; (**D**,**E**) the virus can enter into the brain through the anatomical connection between the meningeal lymphatic vessels and the lymphatic vessels localized along the olfactory nerve [141]. The lymphatic window into the brain is considered in [146,147,148]; (**F**–**H**) The SARS-CoV-2 via the ACE2 receptor on the endothelial cells of the lymphatic vessels [149] can enter into the brain tissues inducing cytokine storm (**F**) and endothelial inflammation that alters the BBB integrity (**G**) facilitating COVID-19-related sleep disorders (**H**). SARS-CoV-2 also affects the olfactory centers (olfactory bulb and cortex), thereby reducing smell sensations [150,151,152,153,154,155,156].

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
