# Peer review of "Brain Mechanisms of COVID-19-Sleep Disorders"

_ijms, 2021, doi:10.3390/ijms22136917_

Round 1

Reviewer 1 Report

The paper is interesting and well written and focuses on an actually topic. 

I suggest to resume in a short table the running results of the paper. 

Author Response

Comments: The paper is interesting and well written and focuses on an actually topic. I suggest to resume in a short table the running results of the paper. 

Response: We would like to thank the reviewer for the valuable advice and support of our article.

We would like to thank the reviewer for the valuable advice and support of our article.

The focus of our review is the analysis of studies related to the blood-brain barrier disruption induced by SARS-CoV-2 as the crucial mechanism responsible for the ‘coronasomnia’ phenomenon. Table 1 summarizes the studies related to the effects of sleep deprivation on the blood-brain barrier functions. We modified Figure 4, which illustrates a brief summary of the most insightful studies referred to our review paper.

We would like to thanks again Reviewer for a great help in improving of our manuscript.                                                                                                               Authors

Reviewer 2 Report

The authors took up a very interesting topic of the Brain mechanisms of COVID-19-sleep disorders. I have some suggestions which may increase the quality of the proposed paper:

- What do we know about the duration of this phenomenon?
- Figure 1 looks strange, it should be deeply reconstructed...
- Lines 111-121 may be discussed with results observed among patients with obstructive sleep apnea: "Currently, there is no biochemical marker useful in the diagnosis of OSA. One of the proposed markers is C-reactive protein (CRP). A meta-analysis revealed a significantly higher level of both CRP and high-sensitivity CRP (hs-CRP) among patients with moderate OSA in comparison with healthy controls.10 However, the AUC calculated for CRP and hs-CRP was under 0.8: 0.73 mg/l (95% CI, 0.60–0.85) and 0.72 mg/l (95% CI, 0.63–0.81), respectively.4,10 These AUC values were lower than those obtained in our study for HIF-1α protein. Similar values to CRP or lower were obtained for glycated hemoglobin, interleukin 6, and erythropoietin.11 " see: https://www.mp.pl/paim/issue/article/15104/
- The paper may benefit from discussing the molecular background of sleep disorders: https://pubmed.ncbi.nlm.nih.gov/33010620/

Author Response

Comment: The authors took up a very interesting topic of the Brain mechanisms of COVID-19-sleep disorders. I have some suggestions which may increase the quality of the proposed paper. What do we know about the duration of this phenomenon?

Response: We want to express our sincere gratitude for great work with our manuscript and deep analysis of our results as well as for constructive and helpful advices for improving of our manuscript.

The most research related to COVID-19-sleep disorders is done on the effect of lookdown or confinement during the pandemic. There is a meta-analysis on the many studies published (https://doi.org/10.5664/jcsm.8930) and once case report (https://doi.org/10.1016/j.nbscr.2020.100057). However, if we want to know the association of a PCR diagnosed COVID-19 infection and sleep problem, there are no studies until now.

Comment: Figure 1 looks strange, it should be deeply reconstructed.

Response: We modified Figure 1.

Comment: Lines 111-121 may be discussed with results observed among patients with obstructive sleep apnea: "Currently, there is no biochemical marker useful in the diagnosis of OSA. One of the proposed markers is C-reactive protein (CRP). A meta-analysis revealed a significantly higher level of both CRP and high-sensitivity CRP (hs-CRP) among patients with moderate OSA in comparison with healthy controls. However, the AUC calculated for CRP and hs-CRP was under 0.8: 0.73 mg/l (95% CI, 0.60–0.85) and 0.72 mg/l (95% CI, 0.63–0.81), respectively.4,10 These AUC values were lower than those obtained in our study for HIF-1α protein. Similar values to CRP or lower were obtained for glycated hemoglobin, interleukin 6, and erythropoietin" see: https://www.mp.pl/paim/issue/article/15104/
The paper may benefit from discussing the molecular background of sleep disorders: https://pubmed.ncbi.nlm.nih.gov/33010620/

Response: We included in the review these recommended articles (Lines 151-156 and 614-621).

We would like to thanks again Reviewer for a great help in improving of our manuscript. We really appreciate you taking the time out to share your experience with us.

Authors

Reviewer 3 Report

The article by Oxana Semyachkina-Glushkovskaya et al. reports an overview of the mechanisms underlying h sleep disturbances associated with COVID-19. The authors explain that at the base of the symptoms of “coronasomnia phenomenon” there are certainly anxiety and stress due to confinement and financial precariousness caused by the pandemic, but in patients affected by COVID-19 sleep disturbances could be caused by molecular events directly triggered by SARS-Cov-2 infection. In fact, the virus can cause, especially in the elderly and debilitated subjects, a cytokine storm that alters the BBB properties, poisoning the brain with repercussions on the quality of sleep. The authors report a summary of sleep deprivation studies in animals showing how insufficient and inadequate sleep increases the inflammatory state and permeability of BBB. It is therefore clear that the inflammation resulting from COVID-19 can facilitate sleep disturbances and in turn a greater brain vulnerability.

Furthermore, the authors suggest that monitoring the sleep status and quality of COVID-19 patients can non-invasively help assess the severity of neurological lesions.

The article is well written and deserves to be published following these minor revisions:

Line 7-8: [email protected] (M.S.), [email protected] (V.A.), 7 [email protected] (E.S.) These names are not included among the authors, so should not be reported next to the Institution name.

Line 86-94-192-447: Please correct “COVID-19 relared” with “related”.

Page 6 (table): “Obstructive sleep apnea is a risk factor triggers neuroinflammation and oxidative-nitrosative stress…”. Please correct “triggers” with “triggering”.

Line 142-143: please correct the sentence “the increased BBB to low molecular-weight tracer sodium fluorescein (10 kDa) and decreased the mRNA levels…” with “the increased BBB permeability to low…..and decreased mRNA level….”

Line 160: please change “the mechanism of sleep loss mediated the BBB opening” with “the mechanism of BBB opening mediated by sleep loss”.

Line 161 : please substitute “increases” with “the increase”.

Line 193-194: please substitute “COVID-19” with SARS-CoV-2.

Line 254: Please add a point after “et al”.

Line 302-303: please first define the full names and then the acronyms in parentheses for ACE2 and TMPRSS2.

Line 314-317: Please provide for MHV, JHM and OMP1 the respective extensive name.

Line 369: please correct “One of hypothesis” with “One hypothesis”.

Line 369-373: The sentence contains too many “and”. Please correct it.

Line 397: MRI should be written in parentheses.

Line 419: Eliminate “is” from the sentence “We raise is the hypothesis…”

Figure 2: Please describe the image on the right. If it is an immunofluorescence image, please give details on the marker(s) detected.

Figure 3: This figure is poorly clear. What is the unit of measure of both x and y axes? Moreover, it is not clear which trend is followed by the molecules named on the graph. Please modify it. Also, the legend should contain the extensive name of every acronym.

Author Response

Comments: The article by Oxana Semyachkina-Glushkovskaya et al. reports an overview of the mechanisms underlying h sleep disturbances associated with COVID-19. The authors explain that at the base of the symptoms of “coronasomnia phenomenon” there are certainly anxiety and stress due to confinement and financial precariousness caused by the pandemic, but in patients affected by COVID-19 sleep disturbances could be caused by molecular events directly triggered by SARS-Cov-2 infection. In fact, the virus can cause, especially in the elderly and debilitated subjects, a cytokine storm that alters the BBB properties, poisoning the brain with repercussions on the quality of sleep. The authors report a summary of sleep deprivation studies in animals showing how insufficient and inadequate sleep increases the inflammatory state and permeability of BBB. It is therefore clear that the inflammation resulting from COVID-19 can facilitate sleep disturbances and in turn a greater brain vulnerability.

Furthermore, the authors suggest that monitoring the sleep status and quality of COVID-19 patients can non-invasively help assess the severity of neurological lesions.

The article is well written and deserves to be published following these minor revisions:

Response: We would like to express our gratitude to Reviewer for the positive and constructive comments. We highly appreciate you taking the time out to share your experience with us.

Comments: Line 7-8: [email protected] (M.S.), [email protected] (V.A.), 7 [email protected] (E.S.) These names are not included among the authors, so should not be reported next to the Institution name.

Response: It was mistake, should be Maria Klimona (M.S.), we corrected it. The other names of Elena Saranceva [email protected] (E.S.) and Vasily Ageev [email protected] (V.A.) were added in the list of co-authors. Elena Saranceva and Vasily Ageev prepared figures for our manuscript.

Comments:

Line 86-94-192-447: Please correct “COVID-19 relared” with “related”.

Page 6 (table): “Obstructive sleep apnea is a risk factor triggers neuroinflammation and oxidative-nitrosative stress…”. Please correct “triggers” with “triggering”.

Line 142-143: please correct the sentence “the increased BBB to low molecular-weight tracer sodium fluorescein (10 kDa) and decreased the mRNA levels…” with “the increased BBB permeability to low…..and decreased mRNA level….”

Line 160: please change “the mechanism of sleep loss mediated the BBB opening” with “the mechanism of BBB opening mediated by sleep loss”.

Line 161 : please substitute “increases” with “the increase”.

Line 193-194: please substitute “COVID-19” with SARS-CoV-2.

Line 254: Please add a point after “et al”.

Line 302-303: please first define the full names and then the acronyms in parentheses for ACE2 and TMPRSS2.

Line 314-317: Please provide for MHV, JHM and OMP1 the respective extensive name.

Line 369: please correct “One of hypothesis” with “One hypothesis”.

Line 369-373: The sentence contains too many “and”. Please correct it.

Line 397: MRI should be written in parentheses.

Line 419: Eliminate “is” from the sentence “We raise is the hypothesis…”

Response: We made the all corrections in accordance with the comments of reviewer.

Comments: Figure 2: Please describe the image on the right. If it is an immunofluorescence image, please give details on the marker(s) detected. Figure 3: This figure is poorly clear. What is the unit of measure of both x and y axes? Moreover, it is not clear which trend is followed by the molecules named on the graph. Please modify it. Also, the legend should contain the extensive name of every acronym.

Response: The figures 2 and 3 were modified.

We would like to thanks again Reviewer for the great support and helpful advices in improving of our manuscript.

Authors

Round 2

Reviewer 2 Report

Authors addressed all the comments improving the manuscript.